# Properties of Cu-xFe_3_O_4_ Nanocomposites for Electrical Application

**DOI:** 10.3390/ma13143086

**Published:** 2020-07-10

**Authors:** Andra Mihaela Predescu, Ruxandra Vidu, Petrică Vizureanu, Andrei Predescu, Ecaterina Matei, Cristian Predescu

**Affiliations:** 1Faculty of Materials Science and Engineering, University POLITEHNICA of Bucharest, RO-060042 Bucharest, Romania; andra.predescu@ecomet.pub.ro (A.M.P.); andrei.predescu@ecomet.pub.ro (A.P.); ecaterina.matei@ecomet.pub.ro (E.M.); predescu@ecomet.pub.ro (C.P.); 2Department of Electrical and Computer Engineering, University of California Davis, Davis, CA 95616, USA; 3Faculty of Materials Science and Engineering, “Gheorghe Asachi” Technical University of Iasi, RO-700050 Iasi, Romania

**Keywords:** nanosize powders, copper, magnetite, powder metallurgy, sintering, thermal properties, magnetic properties, cold-rolling, Vickers micro hardness

## Abstract

Copper matrix nanocomposites reinforced with magnetite nanoparticles were developed using powder metallurgy. Various processing parameters were taken into consideration, such as magnetite content, compaction pressure, sintering time and temperature. The nanopowder blends were compacted using various uniaxial pressures and sintered at 650 and 800 °C in order to study the influence of processing parameters on morphology, structure, thermal, magnetic and mechanical properties. The structure and morphology of the nanocomposites analyzed by X-ray diffraction (XRD), bright field transmission electron microscopy (TEMBF) and scanning electron microscopy (SEM) showed that sintered composites retained the nanoscale characteristics of the initial Fe_3_O_4_ and Cu nanopowders. These nanocomposites have good cold-rolling deformability and Vickers micro-hardness. The Cu-xFe_3_O_4_ nanocomposites have thermal and magnetic properties that make them suitable for electronical applications.

## 1. Introduction

The growing demand for parts with completely new properties arising from conflicting properties has led researchers to design and develop unique composites with bimodal micro and nano-structure [1,2,3]. Nanomaterials are the materials of choice due to their size-dependent properties. The unique properties of nanomaterials open a large spectrum of applications in all of the industrial fields (e.g., nanoelectronics, energy, robotics, rail transportation) but also in medicine, the pharmaceutical industry, environmental fields, engineering etc., where nanomaterial integration technology requires redesign of traditional processes. Examples include dealloying of Al-Cu-Fe quasicrystal alloy in NaOH solution to obtain Fe_3_O_4_/CuO/Cu composites [4] for electrochemical energy storage or sensing devices. Dealloying of FeCuAl alloy is based on the selective etching of Al and spontaneous oxidation of Fe in NaOH solution at room temperature. Tayebi et al. [5] obtained novel Al/B_4_C composites by hot press. They showed that by increasing the amount of reinforcement particles to a percentage of 25%, the coefficient of thermal expansion decreased to a value lower than that of the metal matrix, which recommends the Al/25%B_4_C composite for thermal applications such as heat sink applications. 

Globally, there is a tendency to increase the use of functional and structural materials, such as dispersion-strengthened copper (DSC), and to improve their properties beyond their chemistry. Reinforced Cu-matrix nanocomposites have received special attention due to their nanometer grain size of the copper matrix, which has greater mechanical properties than microcrystalline copper. These nanocomposites could be used as electrical contact materials, where high electrical and thermal conductivity and high mechanical strength at room and elevated temperatures are required [6]. At high temperatures, there is the risk of destabilizing the nanometric structure of these metal oxide composites, which can be prevented by dispersing oxides as a reinforcement phase into the copper matrix to increase creep resistance. The efficiency of nanostructure stabilization is given by the volume fraction of the phases and the degree of dispersion. Such methods for strengthening the matrix by dispersion and improving high-temperature stability has been demonstrated for Cu matrix nanocomposite reinforced with oxide particles such as Y_2_O_3_ [7], ZrO_2_ [8], Al_2_O_3_ [9] and Fe_3_O_4_ [2,10,11]. Copper matrix reinforced with yttrium oxides or aluminum matrix reinforced with Fe_3_O_4_, are used efficiently in aviation and electronics. There is little information in the literature about the Cu-Fe_3_O_4_ nanocomposites, their synthesis and characterization. Previous data on Cu matrix reinforced with magnetite were obtained by direct or reaction milling [11] and powder metallurgy [12]. Cu-based materials obtained by powder metallurgy have good heat conductivity and high anti-wear property, making them suitable for a wide range as aircraft brake materials [12].

In our group, Cu-Fe_3_O_4_ nanocomposites were synthesized by powder metallurgy and a comprehensive characterization of the synthesized Cu and Fe_3_O_4_ nanopowders, and powder blend/mixture during compaction and sintering was performed [2]. We found that the electrical behavior of Cu-xFe_3_O_4_ nanocomposites depends on the amount of reinforcing magnetite; i.e., semiconductors or electric conductors can be obtained. In alternating current, Cu-xFe_3_O_4_ nanocomposites are good resistors too. Depending on the Fe_3_O_4_ content, interesting magnetic properties can be expected by controlling the composition of Cu-xFe_3_O_4_ nanocomposites with bimodal structure. In this work, we extended the characterization of Cu-xFe_3_O_4_ nanocomposites to understand the influence of powder metallurgy (PM) processing parameters such as compaction pressure, sintering time and temperature on the structural and compositional characteristics of the composite, as well as on the thermal, mechanical and magnetic properties of Cu-Fe_3_O_4_ nanocomposites. This research was carried out to obtain advanced nanocomposites with bimodal structure and controlled composition, which possess both electrical and magnetic properties stable at high temperatures. These materials with bimodal structure can find applications in electrical applications.

## 2. Materials

### 2.1. Nanocomposite Synthesis

Nanocomposites were obtained from individual Cu and Fe_3_O_4_ nanopowders with various composition ratios. The synthesis of Cu and Fe_3_O_4_ nanopowders was presented in [2]. Briefly, Fe_3_O_4_ nanopowders were obtained by coprecipitation of ferric and ferrous chloride in the presence of sodium hydroxide, and the copper powder was obtained using the polyol reduction process in 1,2 propanediol. All reagents are of analytical grade and used without further purification from Merck KGaA (Darmstadt, Germany), such as iron(II) chloride tetrahydrate (FeCl_2_.4H_2_O), iron(III) chloride, anhydrous (FeCl_3_), sodium hydroxide (NaOH), urea (CH_4_N_2_O); ammoniac solution, 25% (NH_4_OH) and from Sigma-Aldrich (St. Louis, MD, USA) sodium dodecyl benzenesulfonate (C_12_H_25_C_6_H_4_SO_3_Na) and D-sorbitol (C_6_H_14_O_6_), ethylic alcohol (C_2_H_6_O_2_).

Nanocomposites with copper matrix reinforced with particles of Fe_3_O_4_ were obtained by powder metallurgy from nanosize copper (35–45 nm) and magnetite (5–10 nm) powders. A detailed presentation of the synthesis and characterization of Cu and Fe_3_O_4_ nanopowders was presented in our previous paper [2]. The blends containing Cu with various amounts of Fe_3_O_4_ (i.e., 5, 10, 15 and 20%) were mixed in a tubular laboratory homogenizer. Then, the powder mixture was compacted using a unidirectional press in a CuAl_10_Fe_4_Ni_4_ non-magnetic alloy die, which was custom cast and built in our lab. Powder compaction was performed at 200, 300, 500 and 700 MPa, at room temperature, followed by sintering in vacuum at 650 and 800 °C for 1 h. Sintering of the Cu-xFe_3_O_4_ compacts was carried out in a CALORIS sintering furnace with an adjustable heating system. The sintering time was varied from 60 to 75 min to observe the influence of sintering time on the nanocomposite properties.

### 2.2. Nanocomposite Characterization

Structural characterization was performed to understand the influence of PM processing parameters on the nanocomposite properties. The structure, composition and morphology of nanopowder compacts and nanocomposites were investigated using scanning electron microscopy (SEM), transmission electron microscopy (TEM), bright field transmission electron microscopy (TEMBF) with a high-resolution transmission electron microscope (HRTEM). The TEM investigation was performed on TECNAI G^2^ F30 S-TWIN (Philips, Netherlands) transmission electron microscope with a linear resolution of 1 Å and a punctual resolution of 1.4 Å. Energy-dispersive X-ray microanalysis was performed with an EDAX detector with a resolution of 133 eV.

Thermal properties were investigated with a Laser Flash Analyzer from Netzsch (2009, Selb, Germany) model LFA457, from room temperature to 900 °C. The thermal diffusivity was measured against a commercial copper gauge used as a reference of known specific heat capacity. Differential Scanning Calorimetry (DSC) and Thermogravimetric (TG) were measured on a SetSys Evolution from SETARAM Instrumentation (Caluire, France).

Cold-forming of Cu-xFe_3_O_4_ nanocomposites was performed on a Mario di Maio LQR 120 (Gerenzano, Italy) reversible quartz rolling laminator with the following characteristics: diameter of the working cylinders: 118/53 mm; sheet length: 120 mm; roll jump: maximum 14 mm; rolling speed: 0–6.3 m/min, actual cold work speed: 3 m/min; maximum rolling force of 17 tf; installed power 4 kW. The tension between rolls was adjusted to improve the thickness across the sample. The microhardness test of nanocomposites was carried out using a metallographic microscope, Epitip from Karl Zeiss Jena, Germany, equipped with a microhardness indenter probe (Hanemann type).

Magnetic properties of Cu-xFe_3_O_4_ nanocomposites were investigated using a VSM Model DMS 880 magnetometer (Digital Measurement Systems Inc., Wallingford, CT, USA).

## 3. Results and Discussion

### 3.1. Synthesis, Powder Metallurgy Processing and Characterization of Cu-xFe_3_O_4_

The samples were processed via powder metallurgy technique. For nanomaterials, powder metallurgy processing is still the preferred technology when retaining the nanoscale characteristics of the parent powders is important for the final product. Compared to the classical metal processing by melting and casting, powder metallurgy processing consists of a series of steps, each of them having a critical influence on the final product. Figure 1 shows the processing steps and variables that have been used for Cu-x Fe_3_O_4_ nanocomposites fabrication and characterization. Samples were obtained from individual Cu and Fe_3_O_4_ nanopowders with different Fe_3_O_4_ content, i.e., 5, 10, 15 and 20 wt.%, and under various processing conditions.

#### 3.1.1. Nanopowder Blend

The SEM and Energy Dispersive Spectroscopy (EDS) investigations performed on nanopowder mixtures were used to evaluate the homogeneity of powder composition. Figure 2 presents the SEM image of Cu-15%Fe_3_O_4_ blend in which Fe_3_O_4_ particle are uniformly distributed in the powder mixture. Elemental mapping showed a uniform distribution of Cu, O and Fe, which indicated a good structural homogeneity of the blend. Similar results were obtained for the other nanopowder compositions.

Powder blends are usually described in terms such as flowability, powder filling and density, which can be further used to calculate the consolidation parameters. The theoretical density, *ρ_m_*, was calculated with the following equation:ρ_m_ = %Cu × ρ_Cu_ + %Fe_3_O_4_ × ρFe_3_O_4_(1)
where the theoretical densities for Cu and Fe_3_O_4_ are 8.94 g/cm^3^ and 5.2 g/cm^3^, respectively [2].

The flowability (flowability) of Cu-Fe_3_O_4_ nanopowder blends is virtually zero as they do not flow freely through the fluorimeter port. The apparent density of the powder mixture was determined under the specific conditions specified in the SR EN ISO 3923-1:2010, for non-flowing powders. The apparent density for Cu and Fe_3_O_4_ nanopowders were 1.29 ± 0.17 g/cm^3^ and 0.77 ± 0.07 g/cm^3^, respectively. In the calculations, average values of the apparent densities were taken for both copper and magnetite. The compactness of the mixtures varies between 13.88 and 13.87% and the apparent density between 11.38 and 12.4%.

Figure 3 presents the variation of apparent density and the compactness of the powder with the Fe_3_O_4_ content in the powder mixture. This graph shows that the apparent density of nanopowder mixtures decreased linearly with the Fe_3_O_4_ content from 12.4 to 11.38%, while the powder compactness was almost constant, as it shows only a slight increase with the Fe_3_O_4_ content in the mixture from 13.87 to 13.88%. Actually, the amount of Fe_3_O_4_ should not influence the compactness of the powders, because the size of both powders is in the nanometric range and the blend shows very good homogeneity. 

#### 3.1.2. Powder Consolidation

Because the Cu-Fe_3_O_4_ nanopowder mixture has magnetic properties due to the presence of magnetite in the mixture, steel molds, which are frequently used in powder presses, could not be used. When steel molds were used, the magnetite particles separated on the walls of the mold. Therefore, to press the magnetic nanopowders, a custom non-magnetic die-cast was made. By using this mold, the surface of the compacts was very smooth while preserving the homogeneity of the powder mixture. The consolidation process of Cu-xFe_3_O_4_ blends took place in a cold-rolled unidirectional press. Inevitably, powder agglomeration takes place when nanopowders are compressed in a die.

The electron microscopy images presented in Figure 4 show nanoparticle agglomerations, which is inevitable in the consolidation of very fine, non-flowing powders. SEM investigations were performed on samples consolidation at different pressures (Figure 4 and Figure 5). To obtain the topography of the compacted samples, we used backscatter electron detector (BSED) to detect the elastically scattered electrons from atoms below the sample surface. Figure 5 shows the SEM images of the Cu-15%Fe_3_O_4_ powder mixture pressed at 500 and 700 MPa. After pressing, a rearrangement of the nanoparticles takes place and a decrease in the porosity was observed in both samples. It is obvious that the powder mixture reorganized during pressing to increase compactness. A partial welding processing resulted in agglomerates in the range of 300–600 nm. At 700 MPa compaction pressures, the gliding of the formed agglomerates and the partial filling of the gaps is more noticeable than at 500 MPa (Figure 5).

In the manufacturing of the parts by powder metallurgy processes, an important technological property of any new powder blend is the compressibility of the powder mixture, as a measure of the volume reduction under uniaxial pressure in a closed die. Knowing the compressibility of the Cu-xFe_3_O_4_ powder, the appropriate compact density at a given pressure can be determined, as well as the decrease in the powder volume due to the applied pressure. Further in processing, a high compact density can reduce the dimensional changes produced during sintering.

The compressibility of Cu-xFe_3_O_4_ powders was determined experimentally as compact density function of the applied pressure. The 3-D charts of the powder porosity, compressibility and density as a function of the Fe_3_O_4_ amount in the mixture and compacting pressure is presented in Figure 6, along with copper nanopowder, which is the matrix of the composites. The compressibility data start from the point of apparent density of Cu, i.e., 1.24 g/cm^3^ and increase asymptotically to the theoretical density.

In the case of Cu powder, the compact density does not exceed 8.0 g/cm^3^ at 700 MPa. Although pure copper has a good plasticity, the compressibility is reduced compared to powders of irregular shapes. The reduced compressibility is due to the large particle surface area (~ 50 m^2^/g), the anosized dimension and spherical shape of the particles, which results in a low-pressure distribution on a reduced number of surfaces in contact between nanoparticles.

Regarding the porosity of the compacts, Figure 6 shows that at constant pressure, the porosity increases with increasing Fe_3_O_4_ content, a maximum value of 58.49% being obtained for a composite containing 20%Fe_3_O_4_ pressed at 300 Mpa. At a constant percentage of Fe_3_O_4_, the porosity decreases with increasing compaction pressure, with a minimum value of 12.75% being obtained for Cu powder pressed at 700 MPa. This value increased rapidly to 20.02% when 5%Fe_3_O_4_ was added to copper nanopowder.

Experiments showed that an increase in Fe_3_O_4_ content in the nanopowder mixture contributes to a decrease in the density of presses and their compaction, together with increased porosity. The maximum compaction value of nanopowders was obtained at a pressure of 700 MPa. Because the die did not hold up under 700 MPa pressure requiring repeated adjustments, the subsequent compact samples were obtained at 500 MPa, when the compaction value of the nanopowders varies between 48.83 and 75.41%, depending on the Fe_3_O_4_ content in the blend. It was found that samples containing 5–10% Fe_3_O_4_, have porosities closer to the copper nanometric powders.

During powder compression, various porosities were observed for compacts containing different amounts of Fe_3_O_4_. SEM microscopy images show that the size of the pressed pores increases in proportion to the increase in Fe_3_O_4_ content in the powder mixture. The pore size of presses containing 5–20% Fe_3_O_4_ is between 1.17 and 4.5 μm, as shown in Figure 6.

#### 3.1.3. Sintering of Cu-xFe_3_O_4_ Compacts

Sintering of the Cu-x Fe_3_O_4_ compacts was carried out according to the sintering temperature–time schedule illustrated in Figure 7. The densification of the Cu-xFe_3_O_4_ composites during the sintering process was investigated taking into account the sintering temperature, Fe_3_O_4_ content and sintering time. The remnant porosity and density of the sintered nanocomposites were used as indicators of system sinterability. Characterization of Fe_3_O_4_ nanocomposites obtained by pressing-sintering consists in determining the structure and physical-mechanical properties such as thermal conductivity, porosity, heating behavior, magnetic properties, cold deformability and Vickers micro-hardness. The structure of Cu-xFe_3_O_4_ nanocomposites was observed by scanning and transmission electron microscopy (SEM and high resolution—HRTEM), and X-ray diffraction.

The X-ray diffraction measurements of the sintered Cu-xFe_3_O_4_ nanocomposites show only peaks for the Cu matrix and secondary quasi-crystalline peaks for the reinforced Fe_3_O_4_ particles. Figure 8 shows the X-ray diffraction of Cu-5% Fe_3_O_4_ nanocomposite sintered at 800 °C. Similar peaks for the Cu and Fe_3_O_4_ phases were obtained for all the samples. However, an additional peak was observed for composites containing more than 10% Fe_3_O_4_, which was associated with the CuFeO_2_ phase, also called delaphosite, formed during sintering at 800 °C [2].

SEM images obtained on Cu-xFe_3_O_4_ nanocomposites confirm a good distribution of the particle reinforcement in the copper matrix. Figure 9 shows that the reinforcement particles have rounded and rectangular shapes and they are distributed almost evenly over the surface of the composite material. SEM analysis of these samples also allows for the EDS analysis to be performed on the surface of the particle agglomerates. It has been found that agglomerates contain both magnetite and copper.

A detailed structural analysis of the nanocomposite samples was performed using high-resolution transmission microscopy. Samples for TEM observation were obtained by sampling small portions of the nanocomposite, which were ionically thinned and then collected on a carbon-coated copper grid. Figure 10 shows the bright field transmission electron microscopy (TEMBF) image of the nanostructure of Cu-15%Fe_3_O_4_ composite. At high resolution, the nanostructure of the composite clearly shows the grains of the cupper matrix and the agglomerates of magnetite nanoparticles incorporated in the copper matrix (Figure 10b). The grain size was also measured, showing that both the grains of the copper matrix and the Fe_3_O_4_ particles that reinforced the grains have nanometric dimensions.

The interface between Cu matrix and reinforcing particles was also observed using scanning electron microscopy. In Figure 11, Fe_3_O_4_ particles can be observed at the boundaries of large Cu grains (e.g., the grain of 26 nm in size, Figure 11a), as well inside the Cu grains (indicated by arrows in Figure 11a). The presence of reinforcement particles inside the Cu crystallites and at the grain boundary is due to intense diffusion processes occurring during sintering. Diffusion at the interface between the matrix and the nanoparticles could also explain the formation of an intermediate diffusion layer containing CuFeO_2_.

The presence of Fe_3_O_4_ nanometric precipitates embedded in the Cu crystallites distorts the crystal lattice. In Figure 11b, it is obvious that there is a distortion of the crystalline lattice. The distortion is shown along the white straight line marked on the image, where the local curvature of the crystalline planes is distorted compared to the straight line. The inverse Fourier transform of the image marked with a white square in Figure 11b clearly shows the distortion of the lattice. The distortion of the crystalline lattice is observed by the curvature of the lattice, indicating the existence of nanocrystalline particles (either magnetite or delaphosite).

##### Influence of Temperature and Fe_3_O_4_ Content on Sinterability

During sintering, microstructural changes occur due to the diffusion on the surface, at the interface and in volume. Under heat and pressure conditions, the kinetic energy and the mobility of atoms increase, which cause them to migrate to low energy sites such as cracks and non-even surface to reach equilibrium positions. Surface diffusion results in smoothing the surface of particles and pores. The shape, size and distribution of the pores are dictated by the complex developments that take place during consolidation-sintering process. In the SEM microscopy images of Figure 12, the changes in the morphology of Cu-5%Fe_3_O_4_ and Cu-15%Fe_3_O_4_ compacted at 500 MPa and sintered at 650 °C is observed, along with the temperature changes observed for Cu-5%Fe_3_O_4_.

During sintering, the diffusion of atoms begins at the surface of particles and pores and takes place at the surface of crystalline grains, especially those with crystal defects induced during compaction. At higher temperatures, diffusion in volume predominates and the nucleation of new crystallization centers occurs, especially in strongly deformed areas. Recrystallization begins followed by the gradual growth of newly formed crystalline grains.

After sintering, the samples contracted by approximately 6.67–10%, depending on the Fe_3_O_4_ content. Samples containing 15–20% Fe_3_O_4_ contracted less than those containing 5% reinforcement nanoparticles. The sintering shrinkage was determined according to ISO 4497-2008. When considering the porosity of a sample compaction with the applied pressure (see Figure 6), it is observed that the pores of the sintered composites have shrunk in size or disappeared completely.

X-ray diffraction analysis also provides additional information to understand the phase composition and crystallinity of sintered composites with different Fe_3_O_4_ content. For example, samples with a Fe_3_O_4_ content of less than 10% that were sintered at 650 °C showed no change in phase composition compared to the initial powders. For samples containing more than 15% Fe_3_O_4_ and sintered at 800 °C, a new phase was observed. The new phase delafossite was detected in a percentage of 1–7%, depending on the concentration of Fe_3_O_4_. The composition of this phase suggests that it occurs at the interface between Cu and Fe_3_O_4_ nanoparticles during sintering at 800 °C.

##### Influence of Powder Properties and Pressures on Sinterability

The particle size and shape of the mixture influence the density of the compact and implicitly the sintering density. The finer the granulation of the powders, the greater their sinterability is.

Table 1 presents data on the technological points of reference copper powders with medium size in the micron (< 40 μm) and nanometric (< 35 nm) range, pressed at 500 MPa. It is found that the sinterability of nanoparticles is higher than micron-size particles. The explanation for this can be found in the mechanism of the sintering process. The high specific surface area of nanopowders (50 m^2^/g) results in multiple paths for the transport of atoms by surface diffusion. At the same time, the small size of powder nanoparticles can lead to a reduced size of crystalline grains, which promotes the transport of material through diffusion to the boundary of crystalline grains during sintering. The spherical shape of nanoparticles of nanopowder mixtures has a negative effect on sinterability. Thus, the nanoscale size and spherical shape of the particulate granules results in a lower compact density than in the case of medium-sized powders. A lower compact density means a wider inner surface that favors the sintering process. Densification on sintering increases as the density of compacts is lower, as confirmed by experimental data.

##### Density and Porosity of the Composite

Figure 13 shows the variation of density and porosity of composites with Fe_3_O_4_ content on samples sintered at 650 °C and 800 °C. The compaction pressure was 500 MPa. It was observed that the density of nanocomposites obtained by sintering at 650 °C, decreases with increasing magnetite content, while increasing porosity. At 5–10% Fe_3_O_4_, the composite porosity does not exceed 25%. When the sintering temperature is 800 °C, the density decreases for nanocomposites containing 5–10% Fe_3_O_4_, then increases with increasing Fe_3_O_4_ concentration. This increase in the density of nanocomposites sintered at 800 °C can be explained by the formation of the new phase (CuFeO_2_), previously observed by X-ray diffraction. Nanocomposites have densities close to the theoretical ones and the remaining porosity is much reduced. Densification indexes for sintering are approximately 0.2, and were calculated based on the following formula:(2)$$H_{s}=\frac{\left(\rho_{s}-\rho_{p}\right)}{\left(\rho_{t}-\rho_{p}\right)}$$
where $\rho_{s}$ is the density of the sintered composite (g/cm^3^), $\rho_{p}$ is the density of the compact (g/cm^3^), and $\rho_{t}$ is the true density of the powder mixture (g/cm^3^).

##### The Influence of Sintering Time on Sinterability

It was found that the variation in the sintering time from 60 to 75 min did not significantly affect the porosity of the nanocomposites. As can be seen from Figure 14a,b, the BSED electron microscopy images obtained for samples pressed at 500 MPa and sintered at 650 °C for 60 min, do not essentially differ from those sintered at 75 min. Therefore, a prolonged sintering time is not necessary because the structural transformations that take place during sintering are completed in one hour.

The structural transformations of copper matrix and nanocomposites with 5% and 15% Fe_3_O_4_ content are presented in Figure 15a–c. A non-homogeneity of sintered copper is observed (Figure 15a), while the nanocomposite with 5% Fe_3_O_4_ has a homogeneous structure, the average size of nanoparticles being 44–82 nm. Nanocomposite with 15% Fe_3_O_4_ has a higher porosity and a structural homogeneity lower than 5% Fe_3_O_4_.

The microscopy images showed that the reinforcement particles are uniformly distributed in the nanocomposite. However, research results show that it is difficult to obtain high density of materials, i.e., small porosities, and strong bond between the particles. Agglomeration of the consolidated nanopowders is still a challenging issue. Nanoparticles are strongly influenced by the Van der Waals attraction forces. These forces determine a temporary distribution of loads on each individual nanoparticle, which can lead to rapid agglomeration of nanoparticles. These agglomerates are very difficult to destroy during pressing and sintering, thus contributing to the formation of intergranular voids and the increase of the residual porosity in nanocomposites. When the sintering temperature is 800 °C, advanced compaction occurs in the nanocomposite and the porosity decreases to 3.5%.

### 3.2. Thermal Properties of Cu-xFe_3_O_4_ Nanocomposites 

#### 3.2.1. Thermal Conductivity of Cu-Fe_3_O_4_ Nanocomposites

Nanocomposites with 5–15% Fe_3_O_4_ and nanostructured Cu compacts were sintered at 650 °C and 800 °C, and compared with commercial Cu. Thermal conductivity was obtained by measuring thermal diffusivity and specific heat against a standard, using the following equation:(3)k=c∗α∗ρ
where *k* is the thermal conductivity (W/(m·K)), *c* is the specific heat of the material (J/(kg·K)), *α* is the thermal diffusivity (m^2^/s), and *ρ* is the density of the material (kg/m^3^).

Figure 16a–c shows the thermal properties of the nanocomposite containing 15% Fe_3_O_4_, sintered at 800 °C, compared to a sample of nanostructured Cu obtained under the same conditions and with commercial Cu.

The Cu sample sintered at 800 °C is nanostructured and has a thermal conductivity of 250 W/m.K measured at room temperature. The thermal conductivity values of the microcrystalline copper are between 386 and 401 W/m.K, depending on the temperature at which the determination was performed [13]. The thermal conductivity of nanostructured Cu sample is lower than that of the composite, which can be attributed to the high porosity of the samples processed by powder metallurgy compared to classic methods based on melting and casting. The nanostructured Cu sample has a porosity of approximately 5%, which negatively influences the thermal conductivity of the material. The Cu-15%Fe_3_O_4_ nanocomposite has a lower thermal diffusivity than the commercial and nanostructured copper for the whole temperature range investigated. This is due to the 11% porosity of the Cu-15%Fe_3_O_4_ nanocomposite. 

The values of the measurements performed in the 25–900 °C temperature range for the Cu- Fe_3_O_4_ nanocomposites sintered at 800 °C are presented in Table 2. An increased content of Fe_3_O_4_ in the nanocomposite causes an increase in porosity (see Figure 6), therefore, its thermal conductivity will decrease. Thermal measurements show that at temperatures above 650 °C, a more pronounced decrease in thermal diffusivity is observed for nanostructure Cu (Figure 16a), which could be explained by the increase in grain size at high sintering temperatures. This sudden decrease in thermal diffusivity can be correlated with the increase in specific heat that was observed in the same temperature range (Figure 16b).

In the literature, the thermal conductivity of metallic materials generally ranges from 8.7 to 458 W/mK [14]. Therefore, the Cu-xFe_3_O_4_ nanocomposites exhibit thermal conductivity corresponding to the metallic materials.

Generally, the thermal conductivity of the nanocomposites decreases with the increase of the porosity of the sintered composite, respectively with the increase of Fe_3_O_4_ amount. However, nanocomposites containing more than 15% reinforcing particles and sintered at 800 °C, have a high thermal conductivity due to the formation of CuFeO_2_ during sintering. Nanocomposites sintered at 650 °C exhibit lower thermal diffusivities due to their higher porosity compared to samples sintered at 800 °C.

#### 3.2.2. Differential Scanning Calorimetry (DSC) and Thermogravimetric Analysis (TGA)

The transformations that the Cu-xFe_3_O_4_ nanocomposite undergoes during heating have been studied by differential scanning calorimetry (DSC) combined with thermogravimetric analysis and dilatometry. The nanocomposite sample was heated up to 1000 °C with a heating rate of 100 °C/min in argon atmosphere as a reducing gas. The DSC-TGA curve obtained is shown in Figure 17. From the DSC thermogram, two exothermic effects are observed at 307.92 and 518.53 °C, corresponding to crystalline transition processes in the nanocomposite mass. An endothermic transformation process begins at 892.68 °C, with a maximum intensity at 956.22 °C. This transformation corresponds to the melting point of the nanocomposite Cu-xFe_3_O_4_.

Thermogravimetric analysis performed on the same apparatus shows the weight changes that occur during the heating of the nanocomposite in a controlled atmosphere. By heating in the temperature range 0–1000 °C, the Cu-xFe_3_O_4_ nanocomposite undergoes changes in weight. Thus, between 300 and 850 °C there is an increase by 13.8% in the weight of the sample, while between 850 and 970 °C the weight of the sample decreases by 6.4%. The gain and loss in the composite weight can be explained by absorption and decomposition processes that occur during heating.

#### 3.2.3. Dilatometry

Dilatometry analysis was performed to measure the shrinkage and expansion of the sintered composites over a controlled temperature regime. The coefficient of thermal expansion (CTE) was obtained for the temperature range from 100 to 650 °C. The thermal expansion behavior gives important information for understanding critical engineering changes that take place during the sintering heating processes, the influence of composite constituents, reaction kinetics, and other important aspects of how materials respond to temperature changes.

Dilatometry analysis was performed on a sample of Cu-5%Fe_3_O_4_ nanocomposite sintered at 650 °C. The dilatometric curve is shown in Figure 18. By heating the composite up to 650 °C, i.e., the temperature chosen to mimic the sintering process, the sample does not undergo any changes. The coefficient of linear expansion of the nanocomposite was calculated using the following formula:(4)$$\alpha =\frac{dL}{L_{0}}\cdot \frac{1}{\Delta T}$$
where *α* is the coefficient of linear expansion, *dL* is the change in length of the sample, *L*_0_ is the initial length of the sample, and *ΔT* is the temperature range. The expansion and contraction of the sample during heating and cooling follows a non-linear change. This behavior can be attributed to the deformations such as dislocations, which were induced in the sample during pressing. The shape of the dilatometry curves presented in Figure 18 suggests that the dislocation recovery can happen at various expansion/contraction locations in contrast to sites where regular expansion/contraction occurs.

The expansion coefficient of Cu-5%Fe_3_O_4_ nanocomposite is equal to 16.64 × 10^−6^·K^−1^, and the value of the linear dilatation coefficient of copper is 17 × 10^−6^ K^−1^. The coefficient of thermal expansion is still high for certain electronic application, but increasing the percentage in reinforcement particles in the metallic matrix can decrease the CTE to acceptable values for applications in electronics. Tayebi et al. [5] showed that by adding 25% B_4_C in Al-matrix composite, the coefficient of thermal expansion dropped to a low value of 8 ppm/C, making it suitable for heat sink applications, as demonstrated by the dilatometry tests.

### 3.3. Cold Working Processing Behavior of Cu-xFe_3_O_4_ Nanocomposites

Plastic deformation of sintered materials generally occurs by varying the volume, density or porosity. At the same time, there are also changes in the size and density of the pores. The purity and chemical composition, the initial matrix structure and the specific pore structure of the sintered material are factors that influence the deformability.

Two samples sintered at 800 °C with low and high Fe_3_O_4_ content, i.e., Cu-5%Fe_3_O_4_ and Cu-15%Fe_3_O_4_ respectively, were subjected to cold rolling that followed the reduction plan presented in Table 3 and Table 4. Pass reduction (*ε_i_,* %) and total reduction (*ε_tot_*, %) were calculated using the following equations:(5)$$\varepsilon_{i}=\frac{h_{i-1}-h_{i}}{h_{i-1}}\times 100$$
(6)$$\varepsilon_{i}=\frac{h_{i-1}-h_{i}}{h_{i-1}}\times 100$$
where *h_i−1_* is the thickness before *i* pass [mm], *h_i_* is the thickness after *i* pass [mm], and *i* is the pass number *i* = 1, 2, 3,…., n. Considering the above mentioned parameters for *i* = 1, the *h_i−1_* is *h_0_*, which is the sample thickness before deformation.

The cold rolling was performed until the first micro-cracks visible in close observation with the naked eye appeared on the surface of the samples. (Figure 19) The maximum reduction was 39.70% and 24.16% for sample containing 5% Fe_3_O_4_ and 15% Fe_3_O_4_, respectively. These values obtained for the total reduction are quite remarkable for a composite material.

The quality of the sample surfaces in the cold-rolling process was generally good; no delamination or material exfoliations were observed. Increasing the content of magnetite in the nanocomposite results in a decrease in plastic deformation of the composite, which is due to the higher porosity of Cu-15%Fe_3_O_4_ as compared to Cu-15%Fe_3_O_4_.

In a composite with bimodal structure, the interface between soft and hard area has a deformation mechanism that depends on which phase has the majority. Yu et al. [15] proposed a new deformation mechanism for a hard metal surrounded by a soft metal, such as Cu, based on the concave protrusions of the hard metal that penetrates into the soft metal during roll bonding technique, especially when the thickness of hard metal foil is much less than that of the soft metal foil. The deformation mechanism of the hard nanoparticles can be affected by the tensile stress along the rolling direction because the Fe_3_O_4_ nanoparticles are compressed between Cu “layers”. The Fe_3_O_4_ nanoparticles have a convex shape that can change to concave as the deformation increases with the number of passes.

The Cu nanostructured samples obtained under the same pressing conditions as the Cu-xFe_3_O_4_ nanocomposite was also subjected to the cold rolling deformation process (Table 5).

It is noted that in the cold rolling process, the nanostructured metal achieved a 95.31% reduction rate without the occurrence of cracks. It showed good surface quality in the cold-rolling process without any material stripping or scuffing. This finding confirms the high plasticity of the nanostructured copper.

### 3.4. Vickers Microhardness of Cu-xFe_3_O_4_ Nanocomposites

The Vickers microhardness test was performed according to the EN ISO 6507 standard. The Vickers diamond indenter probe has a square pyramid tip. After pressing the Vickers indenter into the surface, the impression left in the resulted indent was analyzed optically to measure the size of the imprint. Figure 20 shows the Vickers microhardness measurements performed in the Cu matrix as well as on the reinforced Fe_3_O_4_ nanoparticle agglomeration. The values of the Vickers micro-hardness are 50 and 486 HV for cooper and magnetite, respectively.

The micro-hardness of the Cu-5%Fe_3_O_4_ and Cu-15%Fe_3_O_4_ nanocomposite samples sintered at 650 °C and 800 °C, respectively, were measured. Figure 21 shows the copper matrix microhardness varies between 100 and 120 HV, which is approximately 2.5 times larger than microcrystalline copper (50 HV). The presence of a higher concentration of reinforcement particles in the sintered samples at 800 °C leads to an increase in nanocomposite microhardness. Also, the microhardness of the copper matrix differs slightly from one measuring point to another, which indicates a high structural homogeneity of the matrix. The homogeneity has a positive impact on the thermal conductivity of the nanocomposite.

### 3.5. Magnetic Properties of Cu-xFe_3_O_4_ Nanocomposites

The magnetization curves of two nanocomposite samples with 5% and 15%Fe_3_O_4_ were measured at room temperature. Figure 22 shows the variation of the magnetization curves with the application of magnetic field.

Using the SIGMOID Boltzmann function to fit the curves in Figure 22, a small hysteresis was observed, which points to the magnetic anisotropy of the samples. Measuring the saturation levels and coercivity from the hysteresis loop, the characteristics of the magnetization curves of the two nanocomposite samples are presented in Table 6 along with a standard sample. The saturation magnetization value of the standard sample is 55 emu/g, lower than that of bulk magnetite, which is 90.7 emu/g [16].

Superparamagnetic behavior is explained by the fact that particles manifest in the external magnetic field, as magnetic entities that interact only with the field and do not interact with each other [17]. Measurements show that the magnetic saturation increases with increasing Fe_3_O_4_ concentration in the nanocomposite. Thus, the sample with 5% Fe_3_O_4_ content shows a magnetic saturation of 2.85 emu/g, while in the sample with 15% Fe_3_O_4_ content, the magnetization increases to 7.32 emu/g, which is almost three times the value obtained for 5% Fe_3_O_4_.

## 4. Conclusions

Structural and morphological characterizations of Cu-xFe_3_O_4_ nanocomposites obtained by powder mixtures showed that the nanocomposites preserved well their nanometric structure of Cu and Fe_3_O_4_ starting powders. Only at high sintering temperature and high Fe_3_O_4_ content could an additional phase, CuFeO_2_, be observed in the composite structure, which is responsible for remnant porosity and certain thermal and mechanical properties. SEM and TEM investigations also proved that both copper and reinforcing magnetic particles have spherical shapes and nanometric dimensions. The chemical composition of the nanocomposites confirmed their purity as determined by EDS. The electron microscopy study showed that Fe_3_O_4_ nanocrystallites are located within the large grains of the copper matrix and at the interface between the matrix and magnetic particles. This demonstrates the existence of intense diffusion processes that have occurred during sintering by the formation of an intermediate diffusion layer. The presence of Fe_3_O_4_ nanocrystallites within Cu crystals was also confirmed by the distortion of the crystalline lattice of the nanocomposite, similar to nanocrystalline precipitations.

A detailed analysis of the technological indicators related to the applied pressure during compaction (apparent density, filling compactness, shaking density, filling porosity, fluidity of nanopowder mixtures) was performed and the theoretical density of nanopowder mixtures was measured. The results showed that the density and compaction decreased with increasing Fe_3_O_4_ content. Density and compaction also increased with increasing compaction pressure. The compression porosity was calculated and showed values between 20.02% and 58.49% depending on the pressing pressure and the Fe_3_O_4_ content. Increasing the Fe_3_O_4_ increases the porosity at the same compaction pressure, while increasing compaction pressure favors the decrease of compaction porosity.

Sintering at 650 °C led to nanocomposite densities of 6.8–4.2 g/cm^3^, depending on the Fe_3_O_4_ content. Increasing the Fe_3_O_4_ content causes the nanocomposite density to decrease, while the porosity of the nanocomposite increases. Sintering at 800 °C produces nanocomposites with higher densities (7.07–7.9 g/cm^3^), which increases as the Fe_3_O_4_ content increases. Similarly, the porosity of nanocomposites decreases significantly, which was explained by the formation of a new compound, CuFeO_2_, at the interface between magnetic nanoparticles and the copper matrix. The residual porosity of the Cu-xFe_3_O_4_ nanocomposites varies between 3.5 and 20%, depending on the compaction pressure and sintering temperature.

The investigation of the thermal properties was performed on Cu and Cu-15%Fe_3_O_4_ nanocomposite sintered at 800 °C. A slight decrease in the diffusivity of nanostructured Cu was observed compared to the standard metal, probably due to a higher porosity of the Cu nanostructured sample. Additionally, a significant decrease in diffusivity in the nanostructured Cu-15%Fe_3_O_4_ was observed over the entire temperature range tested. At temperatures higher than 650 °C, there was a more significant decrease in diffusivity, which could be correlated with the grain growth at high temperatures. The Cu-xFe_3_O_4_ nanocomposites have thermal conductivities between 83 and 100 W/m.K, which is similar to metals. The thermal conductivity decreases with increasing composite porosity and Fe_3_O_4_ content. For composites containing CuFeO_2_ (i.e., sintered at 800 °C), thermal conductivity increases with increasing Fe_3_O_4_ content. Dilatometry data showed that addition of Fe_3_O_4_ to Cu matrix decreases the coefficient of expansion.

The behavior of the composite under cold rolling showed that the maximum reduction that a nanocomposite can withstand is conditioned by the content of the reinforcement particles. Thus, the first cracks appeared after 39.70% and 24.16% deformation for the nanocomposite with 5% and 15% Fe_3_O_4_, respectively. The Vickers micro-hardness of Cu-xFe_3_O_4_ nanocomposites depends on the Fe_3_O_4_ content in the matrix and the sintering temperature. For the samples containing high Fe_3_O_4_ and sintered at 800 °C, the Vickers microhardness of the copper matrix is about 2.5 times higher than copper.

Magnetization measurements have yielded interesting results. The saturation magnetization of the Cu-xFe_3_O_4_ nanocomposites is 2.8–7.3 emu/g, depending on the amount of magnetite in the sample. The narrow hysteresis loop suggests that a very small amount of energy is dissipated in the revers magnetization, which recommends these composites in alternating current electrical applications such as transformers and motor cores, to minimize energy dissipation with alternating fields.

## Figures and Tables

**Figure 1 materials-13-03086-f001:**
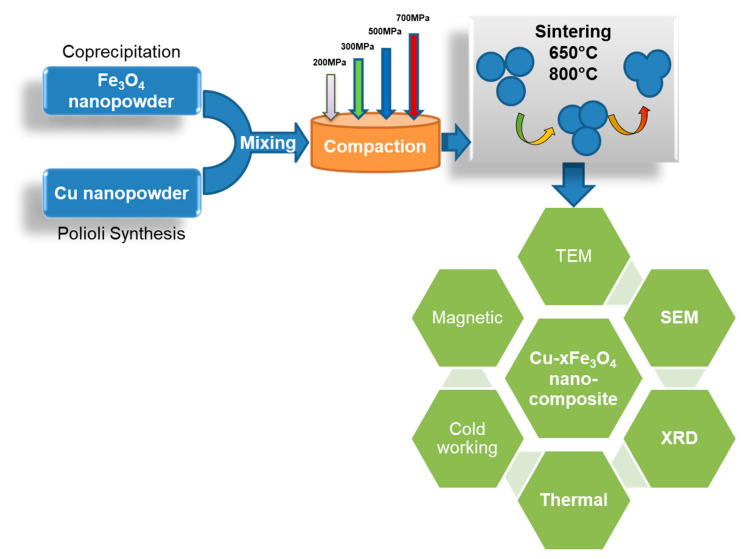
Illustration of the process diagram for synthesis and characterization of Cu-xFe_3_O_4_ nanocomposites.

**Figure 2 materials-13-03086-f002:**
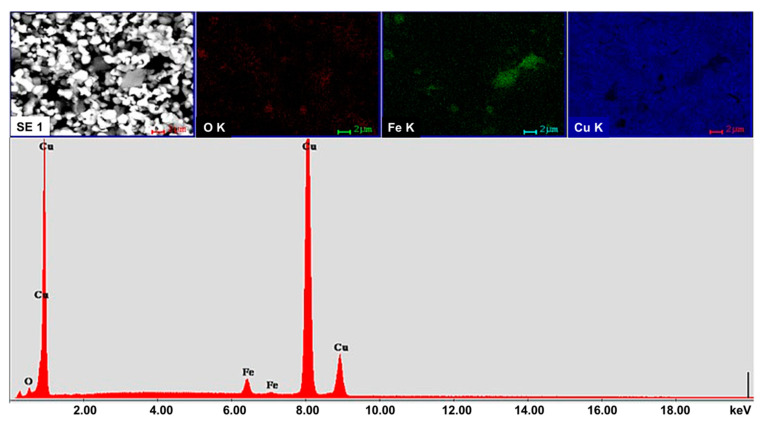
SEM image and the associated elemental mappings of Cu, O and Fe, along with the EDS spectra for a powder mixture of Cu-15%Fe_3_O_4_ (line bar is 2 µm).

**Figure 3 materials-13-03086-f003:**
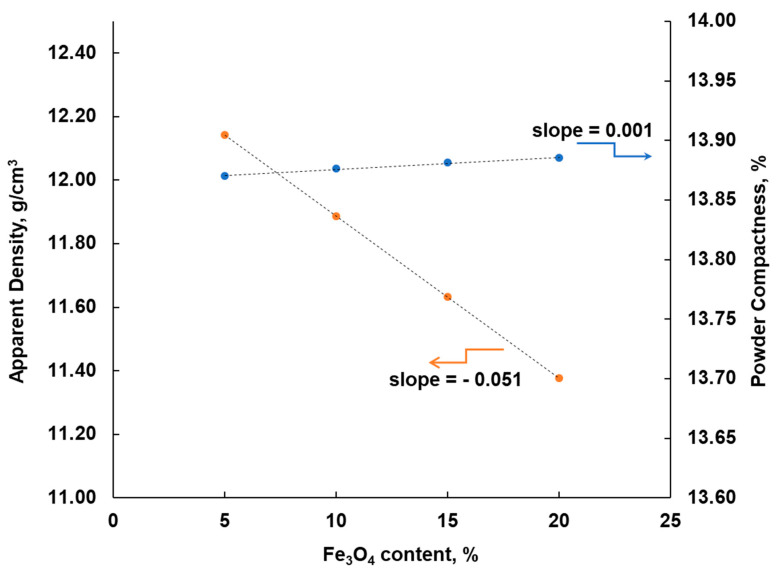
Variation of apparent density and powder compactness with the Fe_3_O_4_ content in the powder blend.

**Figure 4 materials-13-03086-f004:**
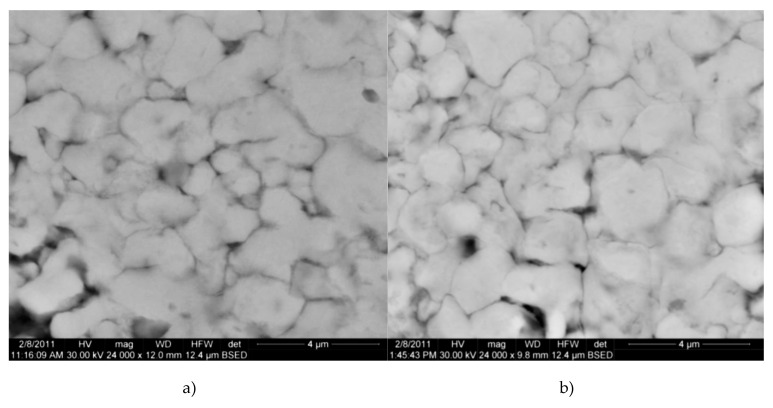
BSED microscopy images of the samples compacted at 500 MPa. (**a**) Cu-15%Fe_3_O_4_ and (**b**) Cu-20%Fe_3_O_4_.

**Figure 5 materials-13-03086-f005:**
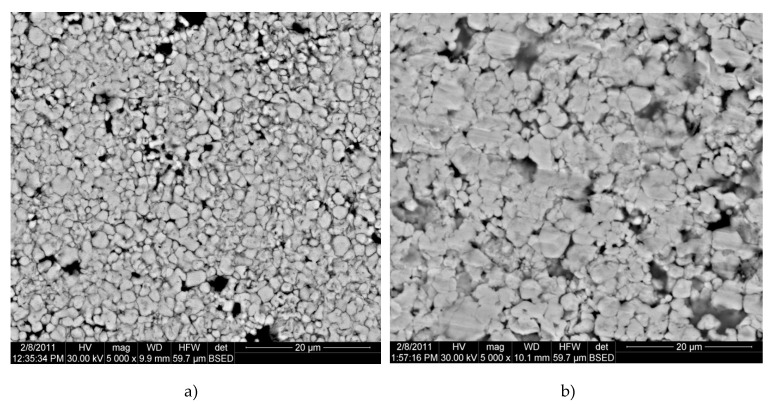
BSED microscopy images of the Cu-15%Fe_3_O_4_ samples compacted at 500 MPa (**a**) and 700 MPa (**b**).

**Figure 6 materials-13-03086-f006:**
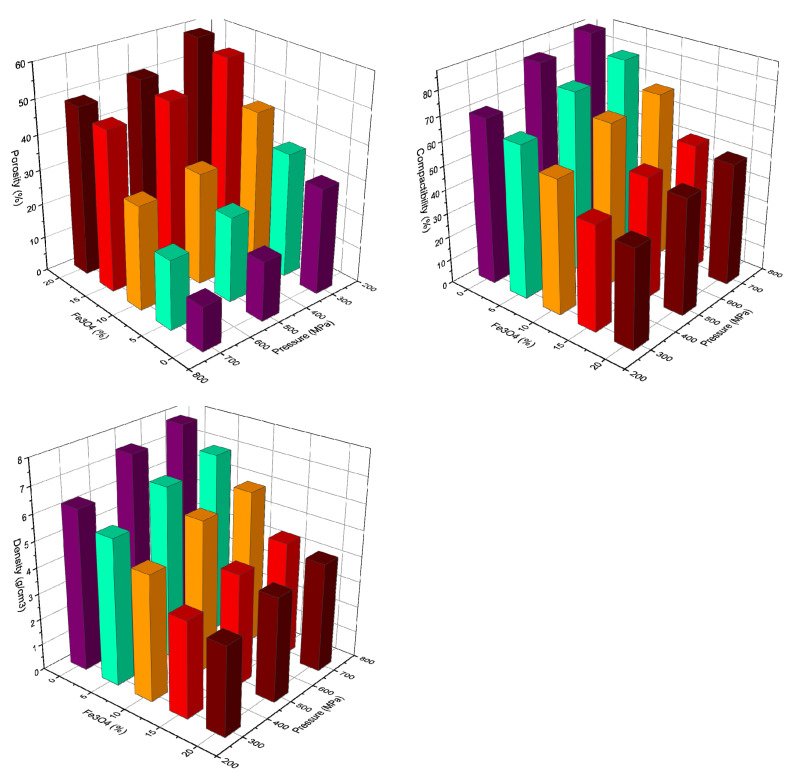
3-D chart representation of the powder porosity, compressibility and density as a function of the Fe_3_O_4_ amount and compacting pressure.

**Figure 7 materials-13-03086-f007:**
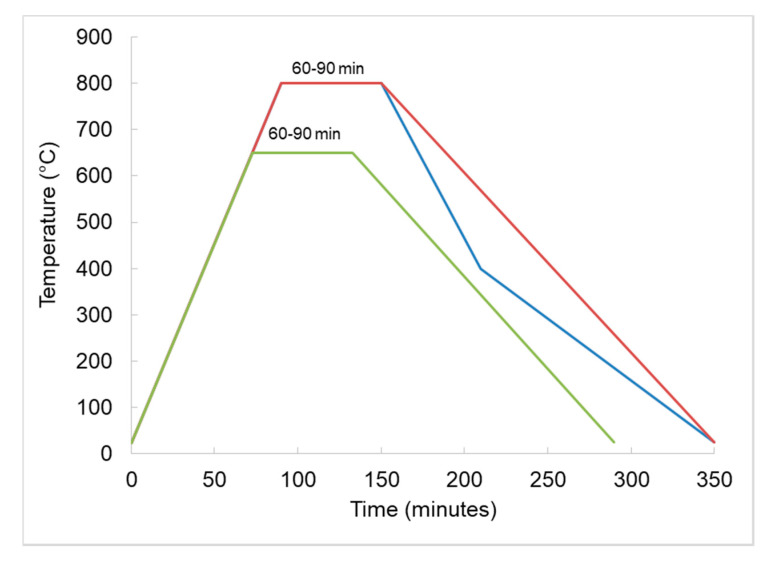
Sintering heating schedule for Cu-xFe_3_O_4_.

**Figure 8 materials-13-03086-f008:**
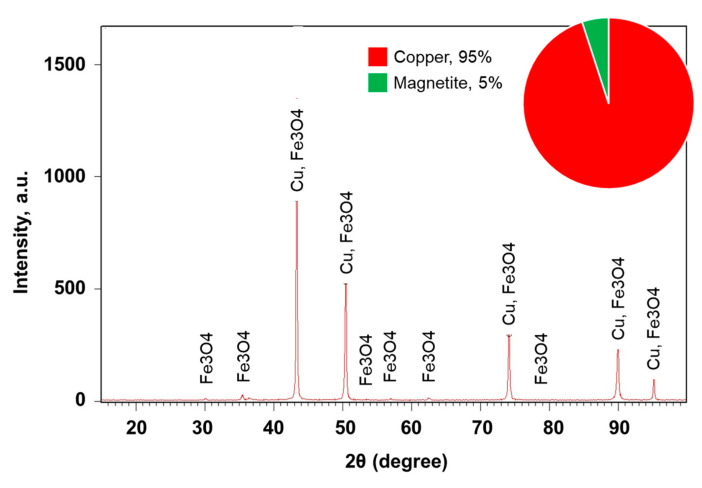
X-ray diffraction of Cu-5%Fe_3_O_4_ nanocomposite sintered at 800 °C.

**Figure 9 materials-13-03086-f009:**
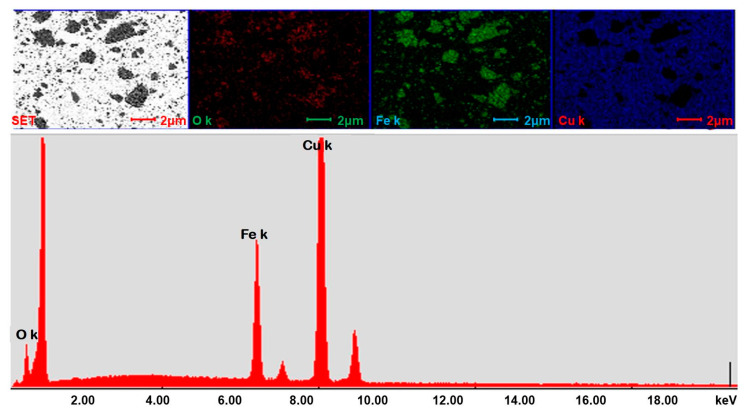
SEM images and EDS spectra recorded for a Cu-Fe_3_O_4_ composite.

**Figure 10 materials-13-03086-f010:**
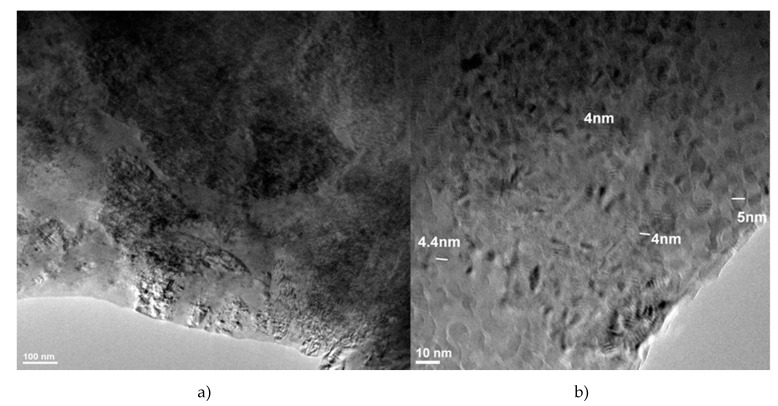
TEM images of Cu-15%Fe_3_O_4_ sintered at 800 °C showing the distribution (**a**) and an agglomeration of Fe_3_O_4_ nanoparticles embedded into the Cu matrix (**b**).

**Figure 11 materials-13-03086-f011:**
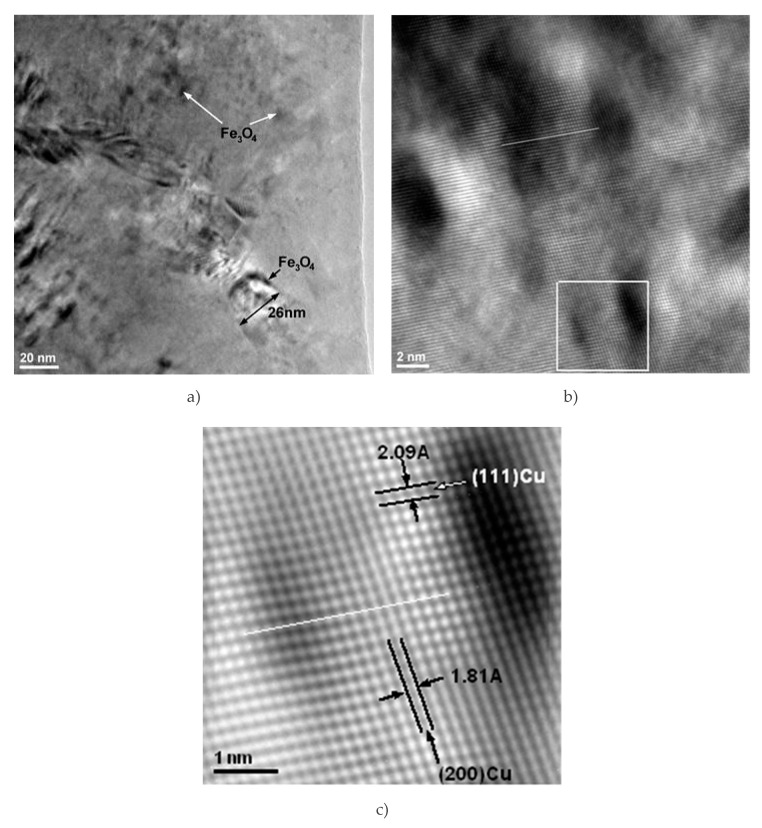
TEMBF images of the interface of Cu matrix and Fe3O4 nanoparticle (**a**); image showing an agglomeration of magnetite nanometer particles embedded in the Cu crystallite matrix (**b**); inverse Fourier Transformation of the image shown in the white square marked in (**b**) at the bottom of the image (**c**).

**Figure 12 materials-13-03086-f012:**
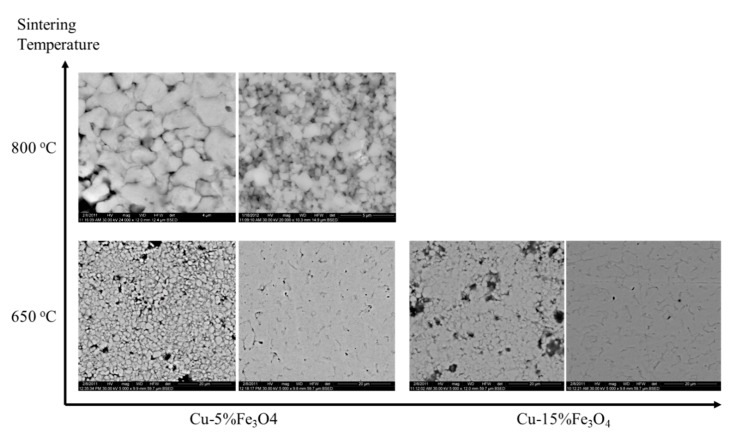
BSED microscopy images of Cu-x%Fe_3_O_4_, where x = 5 and 15; each sintering condition has two images: before (left) and after (right) sintering.

**Figure 13 materials-13-03086-f013:**
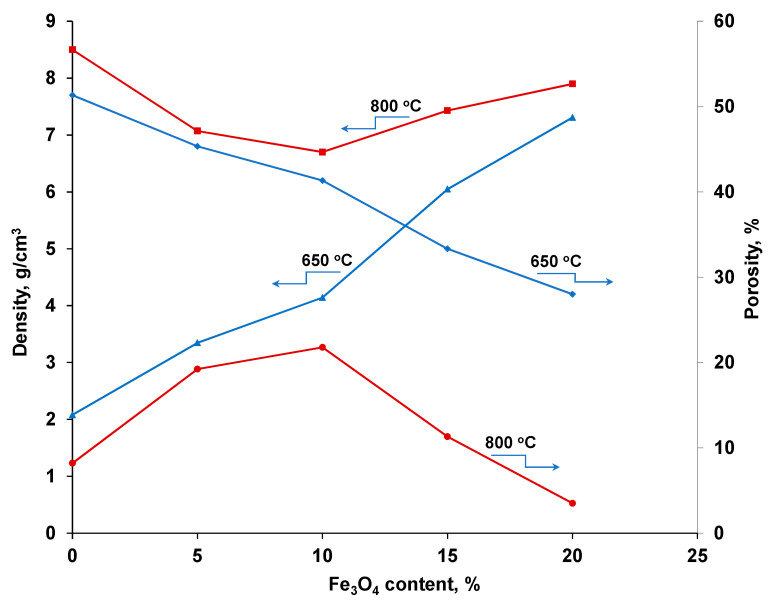
Variation of the density and porosity of composites as a function of the Fe_3_O_4_ content.

**Figure 14 materials-13-03086-f014:**
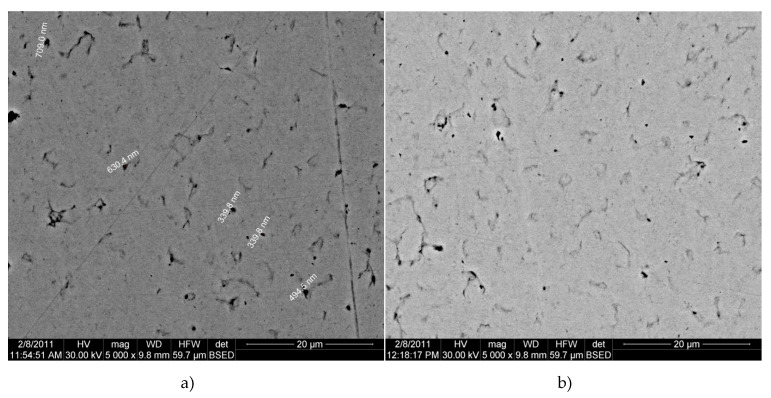
BSED microscopy images of Cu-15%Fe_3_O_4_ nanocomposite obtained by sintering at 650 °C for 60 min (**a**) and 75 min (**b**).

**Figure 15 materials-13-03086-f015:**
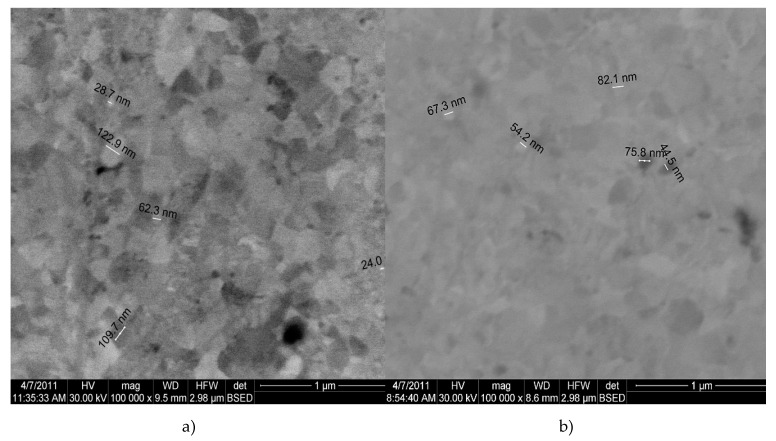
BSED microscopy image of composites sintered at 800 °C: (**a**) Cu; (**b**) Cu-5%Fe3O4 nanocomposite; and (**c**) Cu-10%Fe3O4 nanocomposite.

**Figure 16 materials-13-03086-f016:**
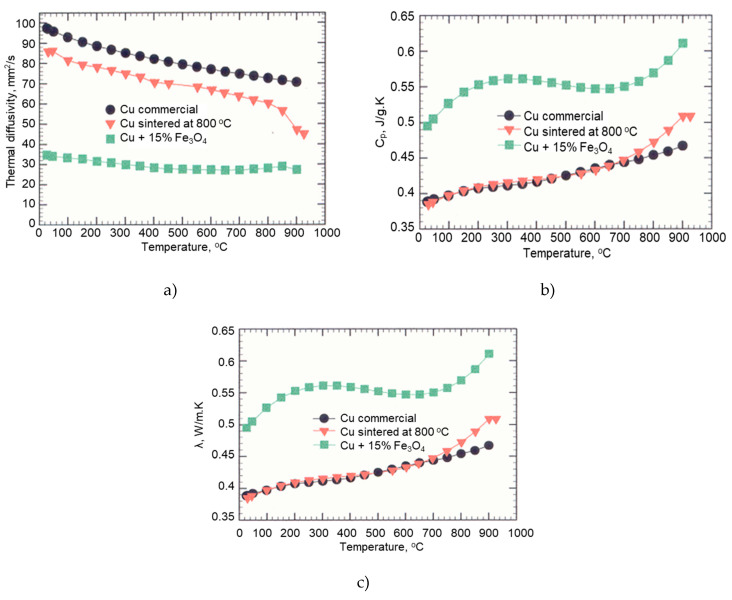
Thermal properties of commercial Cu, nanocomposites of Cu and Cu-15%Fe_3_O_4_ sintered at 800 °C: (**a**) thermal diffusivity; (**b**) specific heat; and (**c**) thermal conductivity.

**Figure 17 materials-13-03086-f017:**
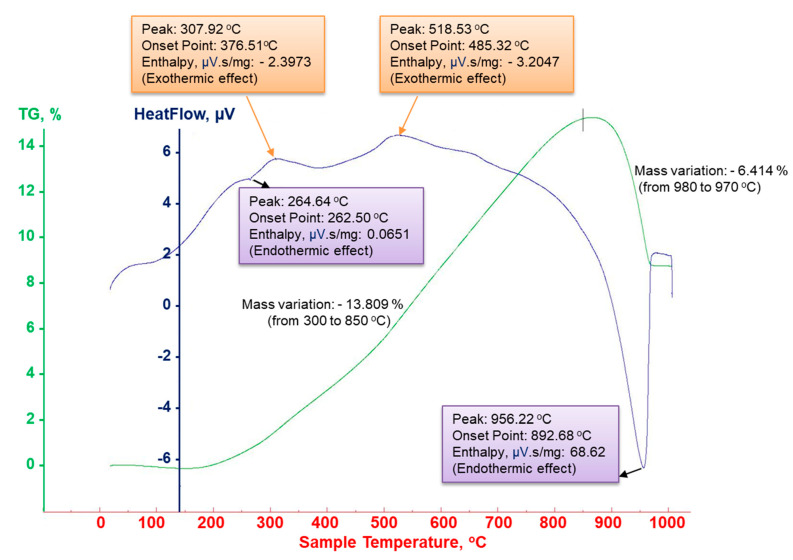
DSC-TG thermogram of Cu-Fe_3_O_4_ nanocomposite.

**Figure 18 materials-13-03086-f018:**
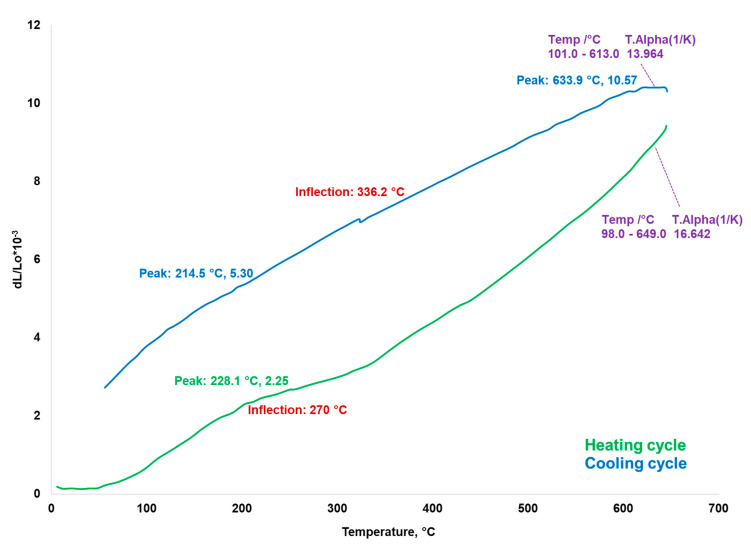
Shrinkage profile as a function of temperature for Cu-5%Fe_3_O_4_ sample.

**Figure 19 materials-13-03086-f019:**
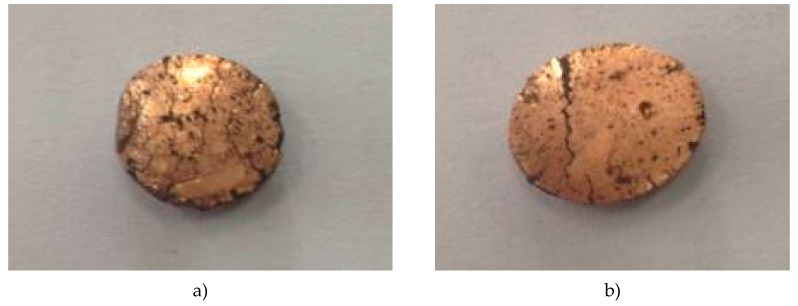
Images of Cu-5%Fe_3_O_4_ (**a**) and Cu-15%Fe_3_O_4_ (**b**) nanocomposites after cold rolling.

**Figure 20 materials-13-03086-f020:**
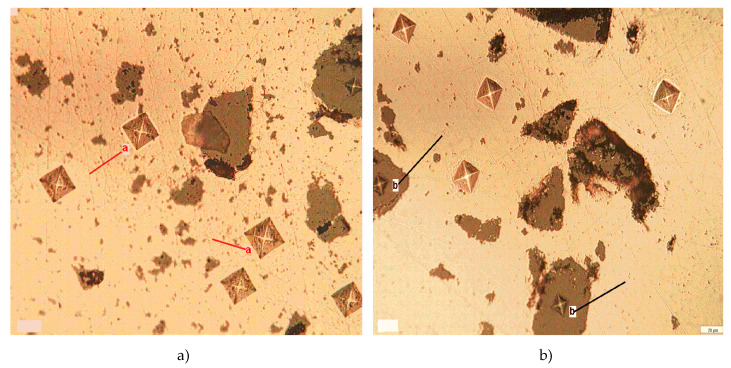
Optical images of the Vickers microhardness measurements on Cu matrix (**a**) and on Fe_3_O_4_ particle agglomeration (**b**).

**Figure 21 materials-13-03086-f021:**
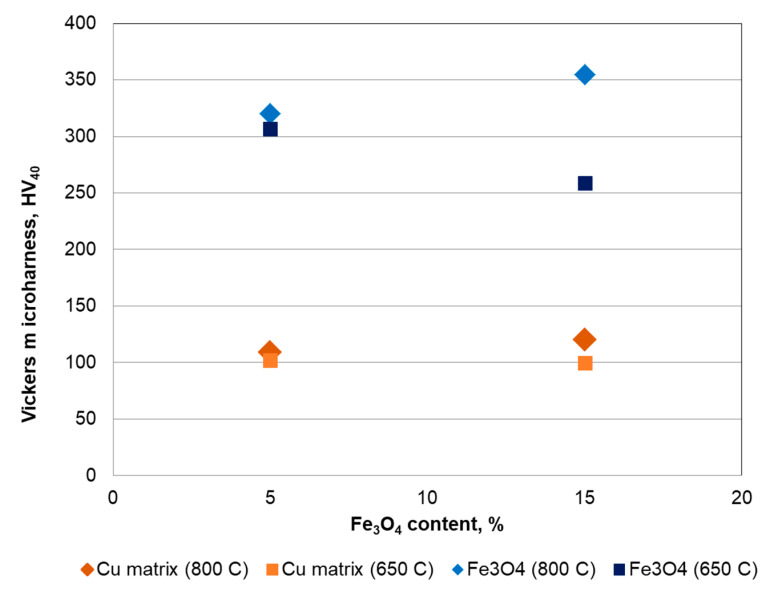
Vickers microhardness (averaged values) of Cu-5%Fe_3_O_4_ sintered at 650 and 800 °C.

**Figure 22 materials-13-03086-f022:**
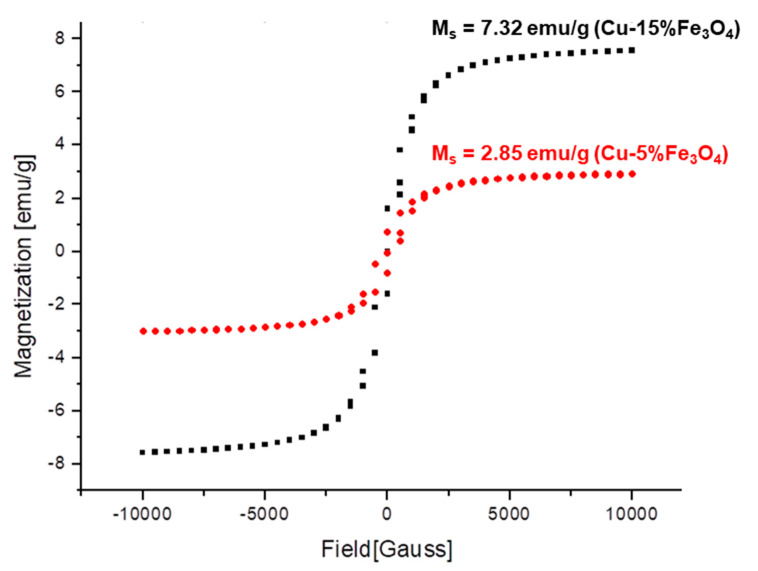
Magnetization curves of Cu-5%Fe_3_O_4_ and 15%Fe_3_O_4_ nanocomposites performed at room temperature.

**Table 1 materials-13-03086-t001:** Technological points of reference for copper powders.

Powder	Compact Density g/cm^3^	Compactness %	Porosity %	
			Compacted	Sintered
Cu, d_m_ < 40 μm	8.20	86.00	14.00	6.00
Cu, d_m_ < 35 nm	7.40	82.80	17.20	4.90

**Table 2 materials-13-03086-t002:** Thermal properties of Cu-15%Fe_3_O_4_ nanocomposites in the 25–900 °C temperature range.

Fe_3_O_4_ %	Sintering Temperature °C	Thermal Diffusivity mm^2^/s	Specific Heat J/g·K	Thermal Conductivity W/m·K
0*	800	89–38	0.38–0.50	250–220
5	800	34–28	0.42–0.40	85–83
15	800	38–30	0.50–0.62	100–90

Note: 0* is for nanostructured Cu.

**Table 3 materials-13-03086-t003:** Cold rolling deformation parameters for Cu-5%Fe_3_O_4_ nanocomposite.

Nr. of Passes	Initial Thickness mm	Final Thickness mm	Pass Reduction %	Total Reduction %
1	6.80	6.35	6.61	6.61
2	6.35	6.05	4.72	11.02
3	6.05	5.84	3.47	14.11
4	5.84	5.74	1.71	15.58
5	5.74	5.56	3.13	18.23
6	5.56	5.40	2.87	20.58
7	5.40	5.20	3.70	23.52
8	5.20	5.00	3.84	26.47
9	5.00	4.75	5.00	30.14
10	4.75	4.53	4.63	33.38
11	4.53	4.33	4.41	36.32
12	4.33	4.10	5.31	39.70

**Table 4 materials-13-03086-t004:** Cold rolling deformation parameters for Cu-15% Fe_3_O_4_ nanocomposite.

Nr. of Passes	Initial Thickness mm	Final Thickness mm	Pass Reduction %	Total Reduction %
1	6.00	5.84	2.66	2.66
2	5.84	5.68	2.73	5.33
3	5.68	5.56	2.11	7.33
4	5.56	5.40	2.87	10.00
5	5.40	5.20	3.70	13.33
6	5.20	4.75	8.65	20.83
7	4.75	4.55	4.21	24.16

**Table 5 materials-13-03086-t005:** Cold rolling deformation parameters for nanostructured Cu.

Nr. of Passes	Initial Thickness mm	Final Thickness mm	Pass Reduction %	Total Reduction %
1	0.64	0.11	82.81	82.81
2	0.11	0.07	36.36	89.06
3	0.07	0.06	14.28	90.62
4	0.06	0.04	33.33	93.75
5	0.04	0.03	25.00	95.31

**Table 6 materials-13-03086-t006:** Magnetic saturation levels and coercivity of the composite samples.

Sample	Magnetic Saturation M_s_, emu/g	Coercive Force Gauss
standard	+55/−55	−5.7/+ 1.71
Cu-5%Fe_3_O_4_	+/−2.85	−187.8/+ 183.9
Cu-15%Fe_3_O_4_	+/−7.3	−287.1/+ 286.2

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
