# Peer review of "Properties of Cu-xFe3O4 Nanocomposites for Electrical Application"

_materials, 2020, doi:10.3390/ma13143086_

Round 1

Reviewer 1 Report

Dear Authors,

I have read manuscript titled: “Properties of Cu/Fe3O4 nanocomposites for electrical application” with great attention.

In my opinion, the article has a good scientific level and can be published after MINOR REVISIONS, because it is an original and valuable work, but I have remarks to manuscript preparation.

Some minor issues to be considered by the Authors:

Please clarify the statement in Line 72 “Interesting magnetic properties” describing the properties of investigating material as “interesting” is very ambiguous.

It is recommended to follow the consistency in notations, I found three different notations for the investigating material: Cu/Fe3O4, Cu-xFe304 and Cu-x%Fe3O4. The reason of using three different notations is not clear and it results in confusion when reading the text.

Please rewrite the sentence in Line 84 avoiding the reference such as “elsewhere”.

Please give the full term before abbreviation in Line 74 “PM”.

Please rewrite the sentence in Line 117, it is not possible to “perform cold plastic deformation”, the plastic deformation might be a result of cold-forming or cold-working.

Please give the full term before abbreviation in Fig. 1 “XRD”.

Please give the full term before abbreviation in Line 140 “EDS”.

In Lines 315-316 it is stated that Fig. 12 embeds the changes in morphology of both Cu-5%Fe3O4 and Cu-15%Fe3O4 sintered at 650 and 800 oC. However, the Fig. 12 is missing the image of Cu-15%Fe3O4 sintered at 800 oC.

Please supplement the adjective “thermogravimetric” in Line 431 with the noun forming full term thermogravimetric analysis (TGA)

Please clarify the difference in initial thickness of Cu-5%Fe3O4 and Cu-15%Fe3O4 in Table 3 and Table 4. Was there any purpose of this discrepancy, because it influence the adequacy of comparison.

Please describe how was obtained/measured the reduction degree in Tables 3 to5.

Please unify and decrease the font size of the figures that it won’t be higher than body text.

Author Response

We thank reviewers for their thoughtful comments on the original version of the manuscript. We have revised the manuscript, accordingly, making the required changes and additions that are highlighted in yellow in the resubmitted manuscript. We agree with the reviewer’s comments that helped improved the clarity of our paper. In the pages below, we will respond in point-by-point fashion to the reviewer’s comments.

I have read manuscript titled: “Properties of Cu/Fe3O4 nanocomposites for electrical application” with great attention. In my opinion, the article has a good scientific level and can be published after MINOR REVISIONS, because it is an original and valuable work, but I have remarks to manuscript preparation.

Some minor issues to be considered by the Authors:

  • Please clarify the statement in Line 72 “Interesting magnetic properties” describing the properties of investigating material as “interesting” is very ambiguous.

R: We modified the sentence as follows: “Depending on the Fe3O4 content, interesting magnetic properties can be expected (...)”

  • It is recommended to follow the consistency in notation, I found three different notations for the investigating material: Cu/Fe3O4, Cu-xFe304 and Cu-x%Fe3O4. The reason of using three different notations is not clear and it results in confusion when reading the text.

R: Per reviewer suggestion, we used Cu-xFe3O4 throughout the text for consistency.

  • Please rewrite the sentence in Line 84 avoiding the reference such as “elsewhere”.

R: We modify the sentence as follows: “The synthesis of Cu and Fe3O4 nanopowders was presented in [2].”

  • Please give the full term before abbreviation in Line 74 “PM”.

R: We modify the sentence to include: “powder metallurgy (PM)”

  • Please rewrite the sentence in Line 117, it is not possible to “perform cold plastic deformation”, the plastic deformation might be a result of cold-forming or cold-working.

R: We modify the sentence to replace “cold plastic deformation” with cold-forming”

  • Please give the full term before abbreviation in Fig. 1 “XRD”.

R: The full term of XRD was given in the abstract.

  • Please give the full term before abbreviation in Line 140 “EDS”.

R: We modify the sentence in Line 140 to include the full term for EDS as: “Energy Dispersive Spectroscopy (EDS)...”

  • In Lines 315-316 it is stated that Fig. 12 embeds the changes in morphology of both Cu-5%Fe3O4 and Cu-15%Fe3O4 sintered at 650 and 800 oC. However, the Fig. 12 is missing the image of Cu-15%Fe3O4 sintered at 800 oC.

R: We modified the sentence to better correlate the text with Figure 12.

  • Please supplement the adjective “thermogravimetric” in Line 431 with the noun forming full term thermogravimetric analysis (TGA)

R: We modified the sentence according to the reviewer comment.

  • Please clarify the difference in initial thickness of Cu-5%Fe3O4 and Cu-15%Fe3O4 in Table 3 and Table 4. Was there any purpose of this discrepancy, because it influences the adequacy of comparison?

R: The difference in the initial thickness of Cu-5%Fe3O4 and Cu-15%Fe3O4 comes from the PM processing of the two powders, which have different densities, compactions and porosities (see Figure 6). The difference in initial thickness should not influence the results, but the reductions, which are given in percentage.

  • Please describe how was obtained/measured the reduction degree in Tables 3 to 5.

R: We have introduced the explanation and the calculation in the text to better describe the values presented in table 3 to 5.

  • Please unify and decrease the font size of the figures that it won’t be higher than body text.

R: Figures have been re-sized to match their fonts with those of the body text. We’ll continue to work with the Editorial office to improve clarity, if needed.

Thank you!

Reviewer 2 Report

In this work, authors developed a magnetite nanoparticles (Fe3O4) reinforced copper matrix nanocomposite using powder metallurgy. Various processing parameters were taken into consideration, such as magnetite content, compaction pressure, sintering time and temperature. This is an interesting and meaningful topic. Therefore, this article can be published on Materials after a minor revision. 1.1. There are too many mistakes in the format of the article, especially the superscript and subscript in the chemical formula. 2.Please pay attention to the sign of temperature (℃). 3.Please improve the quality of the pictures in the article. The SEM images are still not clear after being magnified by 5 times. 4.The author considered many factors, but did not give a clear explanation for each factor. Please optimize the framework of the article. 5.The author should make a summary at the end of each part, not in the conclusion. 6.The title of the article is "Properties of Cu/Fe3O4 nanocomposites for electrical application", but the electrical application is rarely introduced in the text. 7.Please unify the format of references, such as the superscript and subscript in the chemical formula.

Author Response

We thank reviewers for their thoughtful comments on the original version of the manuscript. We have revised the manuscript, accordingly, making the required changes and additions that are highlighted in yellow in the resubmitted manuscript. We agree with the reviewer’s comments that helped improved the clarity of our paper. In the pages below, we will respond in point-by-point fashion to the reviewer’s comments.

Reviewer 2: In this work, authors developed a magnetite nanoparticles (Fe3O4) reinforced copper matrix nanocomposite using powder metallurgy. Various processing parameters were taken into consideration, such as magnetite content, compaction pressure, sintering time and temperature. This is an interesting and meaningful topic. Therefore, this article can be published on Materials after a minor revision.

1. There are too many mistakes in the format of the article, especially the superscript and subscript in the chemical formula.

R: We fixed the format of the chemical formula and temperature unit.

2. Please pay attention to the sign of temperature (℃).

R: We fixed the signed of the temperature.

3. Please improve the quality of the pictures in the article. The SEM images are still not clear after being magnified by 5 times.

R: These images can be re-sized. We will continue to work with the Editorial team to improve their clarity if needed.

4. The author considered many factors, but did not give a clear explanation for each factor. Please optimize the framework of the article.

R: The outline of the article and its framework have been re-evaluated for consistency and clarity. The reviewer may find improved explanations for certain factors. Changes are marked with yellow in the text of the manuscript.

5. The author should make a summary at the end of each part, not in the conclusion.

R: The discussions we performed under each part are relatively short, which also comprise conclusions of each part.  Additional summarization of the results in the form of conclusions would increase the length of the paper making it difficult for readers to follow the correlation between material structure and composition, and properties.

6. The title of the article is "Properties of Cu/Fe3O4 nanocomposites for electrical application", but the electrical application is rarely introduced in the text.

R: The application of this novel material is indirectly reflected in properties. In electrical applications, the material should exhibit good thermal properties and mechanical strength. For certain applications such as transformers and motor cores, magnetic properties are required along with good thermal stability and mechanical strength. 

7. Please unify the format of references, such as the superscript and subscript in the chemical formula.

R: We have formatted the references accordingly. We used EndNote to track references but it doesn’t list the cited references in a correct manner each time. Sometimes, small adjustments are needed when working with Endnote.

We thank reviewer for his thoughtful comments that helped improve the clarity of our paper.